# Experimental quantum compressed sensing for a seven-qubit system

C.A. Riofrío[1], D. Gross[2,3], S.T. Flammia[3], T. Monz[4], D. Nigg[4], R. Blatt[4,5] & J. Eisert[1]

Well-controlled quantum devices with their increasing system size face a new roadblock hindering further development of quantum technologies. The effort of quantum tomography—the reconstruction of states and processes of a quantum device—scales unfavourably: state-of-the-art systems can no longer be characterized. Quantum compressed sensing mitigates this problem by reconstructing states from incomplete data. Here we present an experimental implementation of compressed tomography of a seven-qubit system—a topological colour code prepared in a trapped ion architecture. We are in the highly incomplete—127 Pauli basis measurement settings—and highly noisy—100 repetitions each—regime. Originally, compressed sensing was advocated for states with few non-zero eigenvalues. We argue that low-rank estimates are appropriate in general since statistical noise enables reliable reconstruction of only the leading eigenvectors. The remaining eigenvectors behave consistently with a random-matrix model that carries no information about the true state.

[1] Dahlem Center for Complex Quantum Systems, Freie Universität Berlin, D-14195 Berlin, Germany. [2] Institute for Theoretical Physics, University of Cologne, D-50937 Cologne, Germany. [3] Centre for Engineered Quantum Systems, School of Physics, The University of Sydney, Sydney, New South Wales, Australia. [4] Institut für Experimentalphysik, Universität Innsbruck, Technikerstrasse 25, A-6020 Innsbruck, Austria. [5] Institut für Quantenoptik und Quanteninformation, Österreichische Akademie der Wissenschaften, Technikerstraße 21a, A-6020 Innsbruck, Austria. Correspondence and requests for materials should be addressed to C.A.R. (email: criofrioa@gmail.com).

Recent years have seen rapid progress in the development of quantum technologies, with precisely controlled quantum systems reaching ever larger system sizes. Specifically, for systems of trapped ions, arrays of tens or more individual ions have been engineered and manipulated in their quantum state[1–4], while architectures such as superconducting qubits[5,6] and neutral atoms[7,8], among many others, are also developing rapidly. These technological and scientific developments have enabled implementations of small-scale quantum simulators[2–4], small measurement-based quantum computations[9], proof-of-principle gate-based quantum computations[1,10–12] and quantum error correction, for example, based on topological colour codes[1].

As a result of this fast development, a new roadblock is of increasing concern: the fact that the Hilbert space dimension scales exponentially means that traditional methods for the experimental characterization of quantum states and processes become infeasible even for intermediate system sizes. This is problematic since such systems are the building blocks for emerging quantum technologies. To mitigate this problem, it has been suggested to use various structural properties of natural quantum systems—for example, high purity, symmetries, sparsity in a known basis or entanglement area laws—to reduce the effort of characterization[13–19]. For the purposes of this work, we refer to a quantum system as being intermediate-sized if it has five to ten physical qubits. This is the range where quantum error correction of one- and two-qubit logical gates becomes possible and full-state reconstruction methods are most useful.

Here we are relying on the technique of 'compressed sensing', which has emerged over the past decade in the field of classical data analysis[20,21]. It is now routinely used to estimate vectors or matrices from incomplete information, with applications in such diverse fields as image processing, seismology, wireless communication and many more[21,22]. Furthermore, compressed sensing for low-rank matrices has been adapted as a tool for quantum system characterization—also referred to as quantum tomography—in a series of works[13,15,23]. A particularly appealing feature of 'quantum compressed sensing' (the use of compressed sensing for quantum tomography) is the fact that there is no need to make any *a priori* assumptions about the true quantum state[15,24] since the validity of the reconstruction can be subsequently verified.

Quantum compressed sensing is most effective on density matrices with quickly decaying eigenvalues. Such a state can be well approximated by a matrix of rank $r$ (much smaller than the dimension $d$ of the Hilbert space) and depends only on $O(rd)$ parameters—significantly fewer than the $d^2$ parameters required in general. In quantum information experiments, the goal is often to prepare a pure state, described by a rank-1 density matrix. Noise effects will typically require one to include more than one eigenvalue to obtain a good approximation of the true state. However, in highly controlled experiments, the number of additional eigenvalues is expected to be small. In this context, the theory of compressed sensing showed for the first time that the reduced number of parameters is reflected in a reduced effort in both measurements and computation required for tomographic reconstruction. Indeed, it has been rigorously proven that an (approximate) rank-$r$ density matrix can be recovered from $O(rd \log^2 d)$ experimentally measured parameters[13]. This performance—close to the absolute lower bound of $O(rd)$—can even be achieved when the eigenbasis is completely unknown[13].

In this work, we demonstrate that this approach is reaching maturity by implementing an experimental reconstruction of the state of a seven-qubit system from an informationally incomplete set of measurements. In fact, we report the first implementation of compressed state reconstruction on a platform of seven trapped ions, which are prepared in a state of a topological colour code[25]. Important steps towards quantum compressed sensing protocols have been implemented[17,26,27]. In particular, previous work[27] has explored quantum state tomography based on expectation values using techniques of compressed sensing in a six-qubit photonic system. Here we continue to push towards much bigger systems by developing more scalable compressed sensing reconstruction tools. In addition, we have used a variety of computationally efficient estimators to achieve recovery in practice, which we revisit when we describe the numerical techniques used in our experiment.

A secondary objective of this work is to argue that compressed tomography, while originally developed for density matrices with a small number of dominating eigenvalues, can also be appropriate in situations where the unknown true density matrix is not, in fact, of low rank. This counterintuitive conclusion follows from the finding that in realistic regimes (for example, incomplete data), the statistical signal-to-noise ratio is such that only the leading eigenvectors of the density matrix can be reliably reconstructed. Indeed, we find that the tail of least-significant eigenvectors behaves in ways consistent with a random-matrix model, which means that reporting more than the first few eigenvectors reveals no information about the true state and thus amounts to overfitting. To make this insight more concrete, we formulate a task that is reminiscent of support identification in compressed sensing—the problem of deciding which eigenspaces should be included in an estimate (see, for example, ref. 28), which we call 'quantum support identification'. We give heuristics for identifying the relevant support, based on comparing the behaviour of the estimate with a random-matrix model. Our findings are consistent with a recent approach that recommends spectral thresholding for statistical reasons[29]; and another that shows that statistical noise in state reconstruction protocols can manifest itself by giving rise to random-matrix-like behaviour of the spectrum of the recovered state[30].

Finally, we observe that the estimators introduced in the context of compressed sensing reconstruct the leading eigenvectors more faithfully than more traditional approaches, at the price of being less faithful to the spectral tail. This suggests that one should employ the former if one is more interested in learning about 'coherent errors' (that is, the way in which the first eigenvector deviates from its target), while the latter are better-suited to analyse 'incoherent noise processes' that drive up the rank.

## Results

**Physical system and data model.** We begin by explaining the physical architecture of trapped ions that serves as the platform for this endeavour. In the considered ion-trap quantum computer, $^{40}\text{Ca}^+$ ions are stored in a linear Paul trap. Each physical qubit is encoded in $S_{1/2}(m = -1/2) = |1\rangle$ and the metastable, excited state corresponding to $D_{5/2}(m = -1/2) = |0\rangle$. Manipulation of the qubit is performed by laser pulses resonant (or close to resonant) to the atomic transitions of $^{40}\text{Ca}^+$. The universal set of quantum gates is implemented using three types of operations: collective operations of the form $\exp(-i(\theta/2)S_\phi)$ with

$$S_\phi = \sum_{l=1}^{L} (\cos(\phi)X_l + \sin(\phi)Y_l), \qquad (1)$$

and entangling operations of the form $\exp(-i(\theta/4)S_\phi^2)$, reflecting the entangling Mølmer–Sørensen interaction[31]. Here $X_l$, $Y_l$ and $Z_l$ are the Pauli operators of qubit $l$, $\theta = \Omega t$ is determined by the Rabi frequency $\Omega$ and laser pulse duration $t > 0$, and $\phi$ is determined by the relative phase between qubit and laser. The third type of operations is generated by single-qubit phase

rotations induced by localized AC-Stark shifts. More details of this experimental set-up are covered in ref. 32.

Within this experimental setting involving $L = 7$ qubits, quantum states have been prepared to the best of the experimental knowledge that, however, is limited by statistical noise and systematic errors. The quantum states are described mathematically by density operators $\rho \in \mathbb{H}_d(\mathbb{C})$ (Hermitian $d \times d$ matrices) for $d = 2^L$ that satisfy $\text{tr}(\rho) = 1$ and $\rho \geq 0$. In all of the experiments, the aim was to prepare a pure state vector contained in the code space, which is a two-dimensional subspace of the Hilbert space of seven qubits spanned by $|\bar{0}\rangle$ and $|\bar{1}\rangle$. Here the state vectors $|\bar{0}\rangle$ and $|\bar{1}\rangle$ span the code space and are joint $+1$ eigenstates of the set of stabilizer operators that define the code. The stabilizer operators are given explicitly in ref. 1. The particular basis for the code space is chosen by picking $|\bar{0}\rangle$ and $|\bar{1}\rangle$ to be the eigenvectors of $Z_1 \otimes \ldots \otimes Z_L$ with eigenvalues $+1$ and $-1$, respectively. The states that the ideal experiment would prepare will be referred to as 'anticipated states' in what follows. Both $|\bar{0}\rangle$ and $|\bar{1}\rangle$ are code words of a Calderbank–Shor–Steane code[33,34] originating from the theory of quantum error correction designed to protect fragile quantum information against unwanted local noise. At the same time, they can be seen as the smallest fully functional instances of a 'topological colour code'[25], which are topological quantum error-correcting codes defined on physical systems supported on two-dimensional lattices.

For each state, a set of $n = 127$ Pauli basis 'measurement settings' is chosen. (An informationally complete set would contain $3^L = 2,187$ settings.) Each measurement setting $j$ is characterized by a choice of a local Pauli matrix

$$W_l^{(j)} \in \{X_l, Y_l, Z_l\}, \quad l = 1, \ldots, L, \quad (2)$$

for each of the $L$ qubits. The $l$th qubit is measured in the eigenbasis of $W_l^{(j)}$. There are two possible outcomes for each qubit, and therefore a total of $2^L$ possible outcomes per experiment. Each specific outcome $k$ is associated with a projection operator

$$P_k^{(j)} = \left| v_k^{(j)} \right\rangle \left\langle v_k^{(j)} \right|, \quad k = 1, \ldots 2^L, \quad (3)$$

where $|v_k^{(j)}\rangle$ is a tensor product of eigenvectors of the $W_l^{(j)}$.

For each measurement setting, the measurement is repeated $m = 100$ times and the statistics of measurement outcomes is recorded. From the relative frequencies of outcomes $k$, the probability $\text{tr}(\rho P_k^{(j)})$ is estimated. Note that because of the relatively small number of repetitions of the measurements per setting, given $2^L$ potential outcomes, many of the possible outcomes will not appear even once. This implies that we have a highly noisy signal.

Let us denote the measurement settings that have been chosen as $V \subset W$, where $W$ is the set of all possible measurement settings. We define the 'sampling operator' $\mathcal{A}: \mathbb{H}_d(\mathbb{C}) \to \mathbb{R}^{nd}$ as

$$\mathcal{A}(\rho) = \left( \text{tr}\left(\rho P_1^{(1)}\right), \text{tr}\left(\rho P_2^{(1)}\right), \ldots, \text{tr}\left(\rho P_d^{(n)}\right) \right), \quad (4)$$

with $n = |V|$ the number of chosen settings. That is, the sampling operator is the linear map that simply returns the list of expectation values of the observables $P_k^{(j)}$ measured in the state $\rho$. The data taken are of the type

$$\mathbf{y} = \mathcal{A}(\rho) + \mathbf{z}(\rho), \quad (5)$$

where the zero-mean random vector $\mathbf{z}(\rho)$ captures the statistical noise. The outcomes for any given basis follow a multinomial distribution, from which one obtains the expression

$$\frac{1}{m} \text{tr}\left(\rho P_k^{(j)}\right)\left(1 - \text{tr}\left(\rho P_k^{(j)}\right)\right), \quad (6)$$

for the second moment of each given component of $\mathbf{y}$.

For completeness, we note that the 'Pauli basis measurements' considered here differ from the 'Pauli correlation measurements' that were the basis of previous works on quantum compressed sensing[13]. Pauli correlation measurements are of the form $\text{tr}(\rho(W_1^{(j)} \otimes \ldots \otimes W_L^{(j)}))$, where again the $W_l^{(j)}$ are Pauli matrices acting on the $l$th qubit. These correlators associate 'one' expectation value with each choice of local Pauli matrices and appear, for example, as syndrome measurements in quantum error correction. As detailed above, the 'basis measurements' yield $2^L$ parameters per choice of local Pauli matrices. This is the number of ways of picking one of the two eigenvectors of each Pauli matrix. Basis measurements, which thus give much more detailed information per setting, appear naturally in the ion-trap architecture used in this work. Moreover, one can recover Pauli correlations from basis measurements via the relationship

$$\text{tr}\left(\rho\left(W_1^{(j)} \otimes \ldots \otimes W_L^{(j)}\right)\right) = \sum_{k=1}^{d} (-1)^{\chi(k)} \text{tr}\left(\rho P_k^{(j)}\right), \quad (7)$$

where $\chi(k)$ denotes the parity of the binary representation of the integer $k$.

**Estimators and state reconstruction.** In statistics, an 'estimator' is a rule for mapping observed data (here, outcomes $\mathbf{y}$) to an estimate for an unknown quantity (here, a density matrix $\rho$). At the heart of the discussion is an estimator that is particularly common in the compressed sensing literature. This is the so-called 'trace norm minimizer' referred here to as TNM: this is an estimator based on trace minimization with a positivity constraint. We solve the following problem

$$\min_X \|X\|_* = \text{tr}(X), \quad \text{s.t.} \quad X \geq 0, \|\mathbf{y} - \mathcal{A}(X)\|_2^2 \leq \epsilon, \quad (8)$$

where $\|\cdot\|_*$ is the nuclear or matrix trace norm and $\epsilon > 0$ is the error level. The trace norm can be shown to be the tightest convex relaxation of rank. Thus, the regularization term both encourages low-rank solutions and, due to convexity, can be minimized efficiently[21,23]. This estimator resembles the Dantzig selector[15]. It is an estimator based on the intuition derived from compressed sensing that under the restricted isometry property[35], the positive semidefinite trace norm minimizer compatible with the data is the actual state[13]. This estimator can be cast as a semidefinite program (SDP), which means that $\mathcal{A}(X)$ has to be computed in matrix form, with all memory requirements that come along with it. For $L = 7$ qubits, as presented in this work, using a SDP solver is still feasible since the number of measurement settings is not too large (about $< 1,000$ on a standard workstation). However, better-scaling algorithms exist for this problem and can readily be used. Note that for this estimator, one has to estimate the error-level parameter $\epsilon > 0$ (ref. 15). In practice, one can recognize two limiting regimes for choosing this parameter: if $\epsilon$ is too small, the optimization problem is infeasible, that is, there is no solution $X$ that satisfies the constraints. If $\epsilon$ is too big, in contrast, one finds a solution that it is biased towards low-rank states, that is, solution $X$ is close to being pure. Here we work in the former regime, that is, we use the smallest $\epsilon$ parameter for which the optimization problem converges.

To complement our study, in addition to the TNM estimator, we use a simple and versatile estimator based on the least squares method, which we refer to as LS. It is a LS estimator with positivity constraint and it solves

$$\min_X \|\mathbf{y} - \mathcal{A}(X)\|_2^2, \quad \text{s.t.} \quad X \geq 0. \quad (9)$$

The positivity constraint on $X$ helps the estimation process, based on the intuition that the set of feasible density operators lies at the intersection of those operators compatible with the data and the

positive cone. In practice, one can perform very good estimation with this estimator, even with informationally incomplete measurements, if the actual state is not too mixed and hence close to the boundary of state space. There is empirical evidence for this observation, which can also be made precise[36]. In fact, the additional positivity constraint in this type of problems renders essentially all estimators equivalent[37].

Formally, both estimators can be realized via algorithms that are efficient in the sense that they can be solved in polynomial time in the size of the input. In fact, they are specifically suited to the regime of intermediate/large quantum systems and comparably little data in which we are interested. However, since our analysis involves hundreds of estimation problems, additional theoretical efforts are required to arrive at an implementation that performs well in practice in the regime of intermediate and large quantum systems for the LS estimator. To achieve this, we here introduce a new specific implementation for LS estimator. In this approach, we parametrize the quantum state $\rho$ as

$$\rho = Q^{\dagger}Q, \tag{10}$$

for some $r \times d$ complex matrix $Q$, where $r$ controls the rank of $\rho$. We then consider

$$\min_{Q} g(Q) = \min_{Q} \left\| \mathbf{y} - \mathcal{A}\left(Q^{\dagger}Q\right) \right\|_{2}^{2}, \tag{11}$$

where $\|\cdot\|_{2}$ is the vector 2 norm.

Using this parameterization of $\rho$ ensures that it is positive semidefinite by construction. The optimization problem itself is then solved using a gradient method. A gradient flow on the basis of $\rho$ directly would introduce negative eigenvalues in every step. In contrast, we optimize over $Q$, using the fact that we can analytically compute the gradient

$$\nabla_{Q} g(Q) = 4Q \mathcal{A}^{\dagger}\left(\mathcal{A}\left(Q^{\dagger}Q\right) - \mathbf{y}\right), \tag{12}$$

of the objective function in equation (11). This way, we dispense with the unnecessary and computationally expensive projection step that would otherwise be needed to enforce positivity.

This simplification significantly improves the computational effort as compared to earlier estimators that made use of an iterative gradient method based on $\rho$. We refer to this gradient method based on a manifestly positive parametrization of states as GRAD. We present details of this algorithm in the methods section. This improvement is of great relevance for the study we carry out in the next section.

**Quantum support identification**. The traditional goal of quantum state tomography is to estimate the true density matrix of the system—that is, the one that would result in the limit of infinitely many measurements, when all statistical uncertainties have vanished (assuming no drift or other systematic errors). We will now argue that in a high-dimensional setting, with limited data, it may be neither possible nor desirable to obtain a complete estimate of the true state.

It is not necessarily desirable, because it is unclear that a high-dimensional matrix would provide either interpretable or actionable information. Consider a typical use case for tomography, where the difference between the anticipated state and the leading eigenvectors encodes useful information about the dominating error sources. The eigenvectors associated with the first few eigenvalues contain the most useful information about noise effects, and based in these inputs an experimentalist can adjust the apparatus to achieve a higher fidelity in future runs. However, it is unclear which action would possibly follow from knowing, say, the exact form of the 100th eigenvector.

At the same time, the data obtained may also not be sufficient to estimate all the parameters of the full density matrix to a sensible accuracy. Indeed, trying to fit too many degrees of freedom to noisy data results in 'overfitting', where the estimate depends strongly on statistical fluctuations and only to a small degree on the true state. To combat this, 'model selection' methods give rules for selecting a lower-dimensional model if the amount and variability of the data do not allow for a reconstruction of the full set of unknown parameters[38].

In the context of quantum state estimation, 'spectral thresholding' has been proposed as a model selection method and theoretically analysed in the regime of informationally complete measurements[29]. Spectral thresholding here means that a lower-dimensional model is selected by setting all eigenvalues of the estimate to zero if they are below a threshold value that depends on the dimension of the Hilbert space and the variance of the individual measurements[29].

Here we propose a new heuristic for selecting which eigenvalues of an estimate to keep and which to discard as not meaningful. While it lacks the rigorous guarantees of ref. 29, it is applicable in more general situations. It is based on a transparent criterion: parameters of an estimated density matrix should not be reported if they behave in ways consistent with a random-matrix model—that is, if they can be explained as resulting from a purely random noise without any signal. The use of random-matrix theory for the purpose of testing the significance of estimated small eigenvalues has recently been discussed in ref. 30. While related to our approach, the goals and methods are distinctly different: the aim of ref. 30 is to make statements about the true rank of an unknown density operator by looking at the properties of its spectrum. Here, in contrast, we want to identify the part of the estimate we can already trust given the data available while looking at the properties of the random eigenvectors of the reconstructed density matrix. Our approach, in addition, does estimate the correct true rank when enough information is provided. See Supplementary Discussion for a detailed description.

Technically, for a given data set $\mathbf{y}^{(1)}$, the spectral decomposition of the positive semidefinite estimate $\mathbf{y}^{(1)} \mapsto f(\mathbf{y}^{(1)})$ can be written as

$$\hat{\rho}^{(1)} := f\left(\mathbf{y}^{(1)}\right) = \sum_{j=1}^{d} \lambda_{j}^{(1)} E_{j}^{(1)}, \tag{13}$$

with decreasingly ordered eigenvalues $\{\lambda_{j}^{(1)}\}$ and corresponding eigenprojections $\{E_{j}^{(1)}\}$, where $f$ denotes the action of the estimator on the data. When insufficient data are taken in an experiment, not all eigenprojections can be characterized equally well. Only for some eigenprojections will one have provided sufficient data. They, concomitantly, will have low uncertainties and thus will be common to different estimates of the same state based on different realizations of the experiment, while the other directions will fluctuate wildly based on the particular data obtained. Generating a different data set $\mathbf{y}^{(2)}$ using the bootstrapping techniques detailed below, we arrive at the estimate $f(\mathbf{y}^{(2)})$ with decomposition

$$\hat{\rho}^{(2)} := f\left(\mathbf{y}^{(2)}\right) = \sum_{j=1}^{d} \lambda_{j}^{(2)} E_{j}^{(2)}. \tag{14}$$

Our figure of merit is based on the Hilbert–Schmidt scalar product of the eigenprojections

$$M_{j}\left(\mathbf{y}^{(1)}, \mathbf{y}^{(2)}\right) = \text{tr}\left(E_{j}^{(1)} E_{j}^{(2)}\right), \tag{15}$$

where $j = 1, \ldots, d$. In the informationally incomplete regime we are in, this quantity will show a strong overlap only between the dominant eigenvectors. For the eigenvectors of the complement, the overlaps resemble the overlap of state vectors chosen randomly from the unitarily invariant Haar measure. In the light of this, the spectral thresholding parameter $k$ is taken to be

$$k := \max\left\{ j : \mathbb{E}\left(M_j\left(\mathbf{y}^{(1)}, \mathbf{y}^{(2)}\right)\right) > e_d \right\}, \qquad (16)$$

in expectation over pairs $\left(\mathbf{y}^{(1)}, \mathbf{y}^{(2)}\right)$, where the threshold $e_d$ is chosen as $e_d = \mathbb{E}(x) + \mathrm{var}(x)^{1/2}$ for the random variable defined as $x(U) = |\langle \psi | U | \psi \rangle|^2$ as overlaps between Haar random state vectors from $\mathbb{C}^d$, where $U$ is a Haar random unitary. Specifically, a random-matrix theory computation (see methods) gives,

$$e_d = \frac{1}{d} + \left( \frac{2}{d(d+1)} - \frac{1}{d^2} \right)^{1/2}. \qquad (17)$$

On the basis of such a significance threshold, for the estimate based on the data, we return the spectrally thresholded state $\rho_k$ with a normalization $c > 0$, where

$$\rho_k = c \sum_{j=1}^{k} \lambda_j^{(1)} E_j^{(1)}. \qquad (18)$$

Let us now define the protocol we follow to provide an estimate that has low enough rank to be compatible with few data and yet avoid overfitting. For this, we review the concept of bootstrapping. We consider two types of bootstrapping: parametric and non-parametric. In parametric bootstrapping, from the reconstructed density matrix, one simulates the experimental measurements (sampled according to the appropriate noise statistics) and for each sample data realization one computes a new estimated density matrix. In non-parametric bootstrapping, however, the measured frequencies are assumed as the true probabilities, which in turn are used to simulate (sample) new data sets that are used, as before, to compute an ensemble of estimated density matrices. In both cases, one uses the ensemble of recovered density matrices to gain confidence on the reconstructed state.

The way we proceed is the following: from the experimentally measured frequencies, we do either parametric or non-parametric bootstrapping to generate an ensemble of estimated density matrices using the GRAD algorithm for the LS estimator. We do it in this way because this particular implementation is able to deal with hundreds of bootstrapped data sets in a reasonably short time. Then, we find their spectral decomposition and order their eigenvalues and eigenvectors in descending order as explained above. Subsequently, for all possible pairs of estimated density matrices, we compute the mean of equation (15) for all $j$. Finally, we report as the rank of the reconstructed state the largest $j$, equation (16), for which the quantity $M_j$ has an overlap greater than the threshold computed in equation (17). The results are shown in Fig. 1c and in Supplementary Discussion, where we also show the performance of this method via extensive numerical simulations.

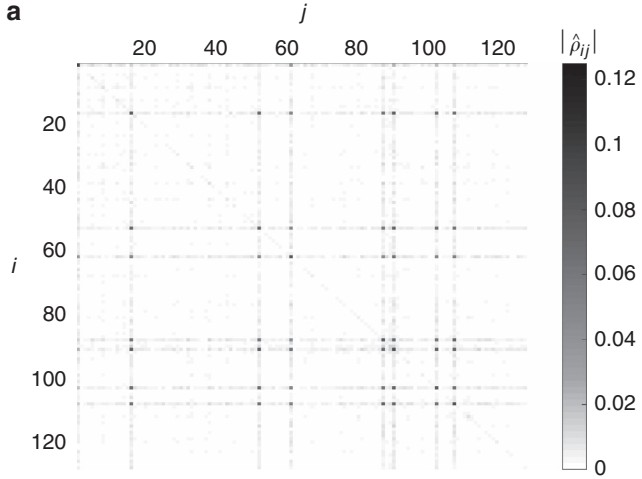

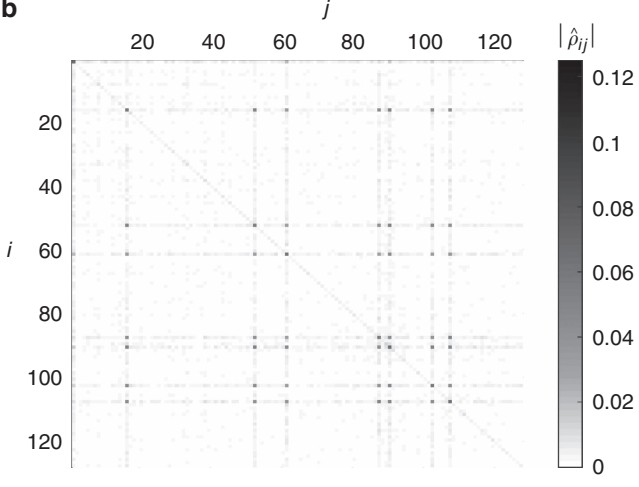

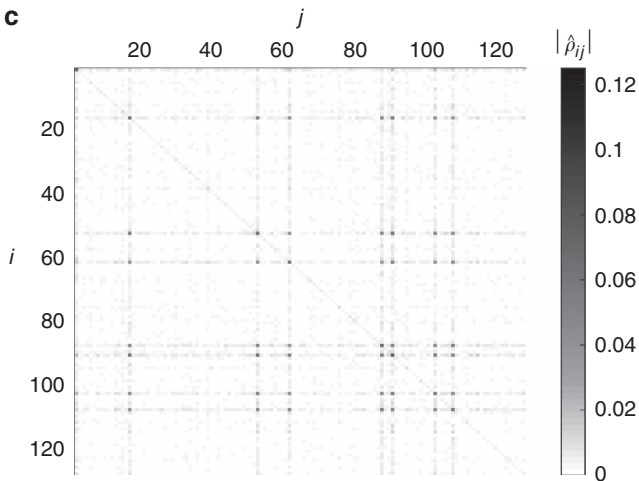

**Figure 1 | Example of quantum state reconstruction for the logical $|\bar{0}\rangle$ state vector.** (**a**) Trace norm minimizer (TNM) estimate with minimal error-level $\epsilon = 1.1$ ($F = 0.43$), corresponding to equation (8). (**b**) Least squares (LS) estimate ($F = 0.30$), corresponding to equation (9). (**c**) Rank 21 leading subspace projection of the LS estimate ($F = 0.32$) obtained by our spectral thresholding method, equation (18). The plots are two-dimensional plots of the absolute values of the entries of the density matrix in the standard basis with magnitude represented by the grey scale. The axes are labelled by the computational basis vectors. For reasons of clarity, the basis vectors are numbered as $x \in \{1, 2, \ldots, d\}$, where $|\chi(x-1)\rangle$ is the state vector in the standard computational basis, and $\chi(x-1)$ is the binary representation of $x - 1$. So $x = 1$ corresponds to $|0, \ldots, 0, 0\rangle$, $x = 2$ to $|0, \ldots, 0, 1\rangle$ and so on. The performance of the reconstruction is measured by the fidelity $F = \langle \bar{0} | \hat{\rho} | \bar{0} \rangle$, where $\hat{\rho}$ is the estimated state. While all three estimators produce roughly similar looking estimates, they differ in the fidelity with the anticipated state for the reasons explained in the main text.

## Discussion

In each experiment we have performed, the 'anticipated states' were taken from the code space $\mathrm{span}(|\bar{0}\rangle, |\bar{1}\rangle)$ of the topological colour code. In Fig. 1, we present a graphical representation of an instance of such a reconstruction (for more examples, please see Supplementary Discussion). It depicts the reconstructed state based on the TNM estimator, the one obtained via the LS estimator computed with our GRAD algorithm, and a truncated estimate that we call 'spectral thresholding' estimate, where only the highest eigenvalues have been kept via the procedure described in the previous section.

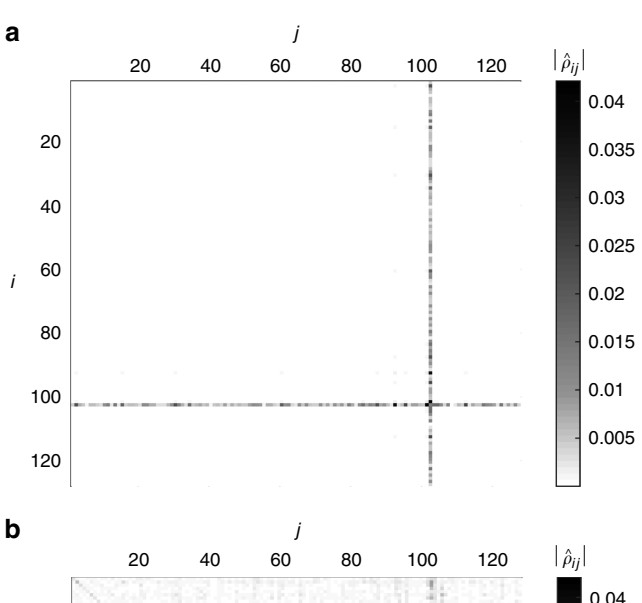

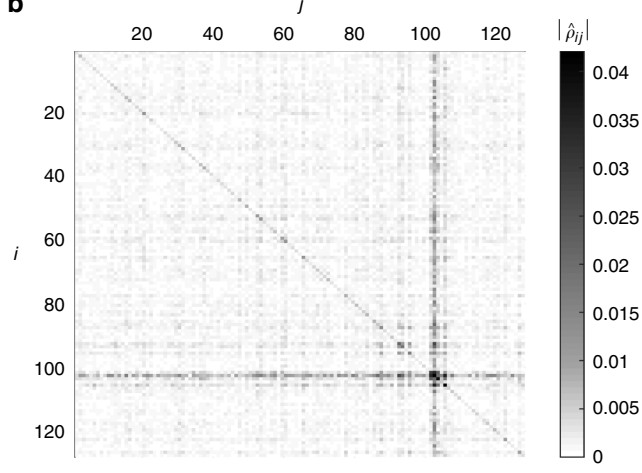

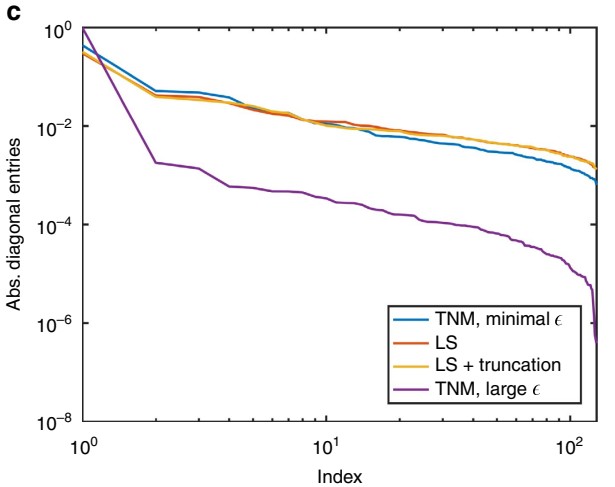

From Fig. 1, we see that the estimators give rise to valid and faithful reconstructions of the anticipated state, in that the reconstructed states are close in fidelity to the anticipated states, though some estimators report higher fidelities than others. The computational runtime for the estimators is moreover quite modest, of the order of an hour, and of a few minutes for the GRAD estimator. This analysis, Fig. 1a, can be seen as a first experimental implementation of quantum state tomography based on a compressed sensing methodology for high-dimensional quantum systems.

If we compare a typical figure of merit for the quality of a state reconstruction—the fidelity to the anticipated state—then we notice that the three reported reconstruction fidelities differ from $F = 0.43$ (TNM estimate) down to $F = 0.30$ (LS estimate) on the same data. We hypothesize that this difference is due to a combination of limited data (applicable to all estimators) and the fact that the TNM estimate gives a much higher penalty to mixed states than the other two estimators. In fact, we empirically observe that the purity of the reconstructed state strongly depends on the choice of error-level parameter $\epsilon$. For large $\epsilon$ values, the TNM estimator will favour reconstructing nearly pure states, and thus it will not estimate the tail of the spectrum when data are scarce. Because of this, it is expected to be better at diagnosing 'coherent' errors, as it will return the dominant pure state. By contrast, for small $\epsilon$ values or when other estimators are used, one sacrifices purity to better match the spectrum, which causes the reconstructed state to have a poorer fidelity with the anticipated state. Thus, these estimators might be better for diagnosing 'incoherent' noise, as these methods retain visibility for excessive noise in the system. In the context of process tomography of near unitary maps, a similar effect has been observed[39].

We can gather evidence for our hypothesis by looking at the diagonal matrix elements of the reconstructed states in the anticipated basis, meaning the stabilizer basis that includes the anticipated state. In Fig. 2a,b, we see the absolute values of the matrix elements of the reconstructed states using this basis. Here it is much clearer that the TNM estimate with large $\epsilon$ parameter is detecting coherent noise, while the LS estimate (and the spectral thresholding estimate, not shown) achieve a more mixed reconstruction. In fact, Fig. 2c shows that in every case the majority of the diagonal elements are decaying exponentially when ordered in decreasing magnitude, but with a much more rapid initial decay for the TNM estimator with

**Figure 2 | Coherent versus incoherent error analysis for the logical $|\bar{0}\rangle$ state vector.** (**a**) Trace norm minimizer (TNM) estimate with large $\epsilon = 1.8$. (**b**) Least squares (LS) estimate. (**c**) Diagonal element comparison. (**a**,**b**) Two-dimensional plots of the absolute values of the entries of the difference between the anticipated state and the reconstructed state density matrices in the stabilizer basis of the anticipated state for the logical $|\bar{0}\rangle$ state vector. In this basis, the anticipated state is exactly diagonal with only one non-zero entry in the diagonal. While only the TNM (with large $\epsilon$) and LS estimates are shown, the spectral thresholding estimate is very similar to (**b**) and is omitted. (**c**) In the same basis, we plot the diagonal elements of the reconstructed density matrices in order of decreasing magnitude. The log–log plot shows that after a rapid initial decay, most of the diagonal elements follow an exponential decay curve. The TNM (with minimal $\epsilon$) has slightly less heavier tails than the LS and spectral thresholding estimates. For comparison, the result of the TNM estimator with a large $\epsilon$ parameter has almost all its support in few diagonal elements and thus is biased heavily towards pure states, as expected. As discussed in the main text, the TNM estimate with large $\epsilon$ parameter is detecting coherent noise, while the LS estimate achieves a more mixed reconstruction and is better-suited to detect incoherent errors.

large $\epsilon$. Although this constitutes evidence for our hypothesis, much more work should be done to determine whether there is any advantage to using different estimators to highlight different features of the noise.

In this context, from our analysis of the experimental data at hand, and given the amount of available data, we see evidence that the errors present in our experiment are mostly incoherent. The absence of strong coherences (off-diagonal elements) in the reconstructed density matrix (Fig. 2a; Supplementary Fig. 3) seems to indicate that. Our methods demonstrate that the principal components of the density matrix have a large overlap with the anticipated state. Notice that observing this feature in such a large Hilbert space by luck is nearly impossible. This observation gives further actionable advice from the tomographic analysis, in that an improvement of the preparation should aim at removing such incoherent errors.

As stated earlier, the purpose of our work is twofold. On one hand, it presents a successful first compressed sensing tomography implementation on a seven-qubit quantum experiment, using estimators and reconstruction techniques that are efficient in the Hilbert space dimension. In this way, it demonstrates the potential of using the machinery of the 'big data' paradigm to assess quantum systems close to the limit of what is experimentally feasible. On the other hand, more conceptually, we discuss ideas of 'quantum support identification', related to the question of what quantum state tomography can actually mean in the regime of informationally incomplete data for intermediately sized quantum systems. We advocate a paradigm that only those low-rank states should be reported that have a statistical basis and we see that the compressed sensing machinery, that is, TNM with minimal $\epsilon$, is enough to give an estimate of such characteristics without being too expensive to compute. We also saw evidence that using a TNM estimate with large $\epsilon$ can be useful to detect coherent errors.

Quantum tomography—the task of reconstructing unknown states from data—is a key primitive in quantum technologies. At its heart, it aims at providing actionable advice upon which the experimenter can make the appropriate modifications to an experimental set-up. It goes beyond mere certification of the correctness of an anticipated preparation of a quantum state[15,16,40]: by learning in what way the actually prepared state deviates from the anticipated one, one can modify the apparatus appropriately to improve performance in future runs. It is our hope that we can inspire further work on the certification and reconstruction of quantum states and processes for increasingly large quantum systems, overcoming the roadblock against further development in quantum technologies.

## Methods

**The GRAD algorithm.** Here we present details of the reconstruction sketched in the main text. In our approach, which we called GRAD, solving the LS estimator, we parametrize the density matrix as

$$\rho = Q^{\dagger}Q, \tag{19}$$

which makes it manifestly positive semidefinite. We then solve

$$\min_{Q} g(Q) = \min_{Q}\left\|\mathbf{y} - \mathcal{A}\left(Q^{\dagger}Q\right)\right\|_{2}^{2}, \tag{20}$$

with $\|\cdot\|_{2}$ being the vector 2 norm. We do this using a gradient search algorithm, but notably for $Q$ and not for $\rho$ itself. This is the key feature of this approach. This method derives from the idea presented in ref. 41. The basic iteration step in a sequence of $\{Q_i\}$ is

$$Q_{i+1} = Q_i - \alpha_i \nabla_Q g(Q_i). \tag{21}$$

For the moment $\alpha_i > 0$ is chosen to be a sufficiently small step size, but this can surely be refined to a conjugate gradient method if absolutely necessary, and can hence be tuned to increase convergence speed. In our case, the actual gradient can

be computed analytically. Note that $g(Q)$ can be written as

$$g(Q) = \sum_k \left(y_k - \mathrm{tr}\left(P_k Q^{\dagger}Q\right)\right)^2, \tag{22}$$

and its gradient

$$\begin{aligned}\nabla_Q g(Q) &= -2\sum_k \left(y_k - \mathrm{tr}\left(P_k Q^{\dagger}Q\right)\right)\nabla_Q \mathrm{tr}\left(QP_k Q^{\dagger}\right) \\ &= 4Q\mathcal{A}^{\dagger}\left(\mathcal{A}\left(Q^{\dagger}Q\right) - \mathbf{y}\right),\end{aligned} \tag{23}$$

as stated in the main text. Here we have used the standard matrix identity $\nabla_X \mathrm{Tr}(XBX^{\dagger}) = XB^{\dagger} + XB$ for the particular case in which $B$ is Hermitian. We then iterate the previous equation until reaching convergence. The state is renormalized at the end, as the trace is not constrained in this way. This is an extremely fast and elegant way to incorporate positivity of $Q^{\dagger}Q$.

It is worth mentioning that this approach significantly improves earlier ideas deriving from refs 42,43, in which a gradient method for the state $\rho$ was combined with a suitable projection. Specifically,

$$\min_X f(X) = \min_X \|\mathbf{y} - \mathcal{A}(X)\|_2^2, \quad \text{s.t.} \quad X \geq 0 \tag{24}$$

was solved by moving away from the SDP and solving the optimization problem using a gradient search algorithm. The basic iteration is

$$X_{i+1} = \mathcal{P}(X_i - \alpha_i \nabla_X f(X_i)), \tag{25}$$

where here $\nabla_X$ is the gradient operator with respect to matrix $X$ and $\mathcal{P}(X)$ is a projector that makes the estimated state positive semidefinite. The gradient is computed explicitly

$$\nabla_X f(X) = 2\mathcal{A}^{\dagger}(\mathcal{A}(X) - \mathbf{y}). \tag{26}$$

While this approach also works, the projection significantly slows down the algorithm, hence the need for our method that directly incorporates positivity.

**The random-matrix model.** Here we present results from random-matrix theory on expected overlaps of random vectors. Specifically, for an arbitrary vector $|\psi\rangle \in \mathbb{C}^d$, we consider the random variable defined by

$$x(U) = |\langle\psi|U|\psi\rangle|^2 \tag{27}$$

and moments thereof with respect to the Haar measure. This quantity is easily identified as the overlap of two random vectors from $\mathbb{C}^d$. We compute first and second moments thereof. They can be computed making use of the powerful Weingarten function formalism[44]. We find in terms of a Weingarten function $Wg$,

$$\int_{U(d)} dU |\langle\psi|U|\psi\rangle|^2 = Wg((1), d) = \frac{1}{d}. \tag{28}$$

The second moments can be expressed as

$$\begin{aligned}\int_{U(d)} dU |\langle\psi|U|\psi\rangle|^4 &= 2Wg((1,2), d) + 2Wg((2,1), d) \\ &= \frac{2}{d^2 - 1} - \frac{2}{d(d^2 - 1)} \\ &= \frac{2}{d(d+1)}\end{aligned} \tag{29}$$

using suitable Weingarten functions. The sum over all permutations on two symbols in the relationship between Haar averages and Weingarten functions, $(1,2) \mapsto (1,2)$ and $(1,2) \mapsto (2,1)$, then simply gives rise to the above two terms. These result in the expression for the variance

$$\mathrm{var}(x) = \frac{2}{d(d+1)} - \frac{1}{d^2}. \tag{30}$$

In the main text, the quantity

$$\begin{aligned}e_d &= \mathbb{E}(x) + \mathrm{var}(x)^{1/2} \\ &= \frac{1}{d} + \left(\frac{2}{d(d+1)} - \frac{1}{d^2}\right)^{1/2}\end{aligned} \tag{31}$$

has been derived from this.

**Data availability.** The data that support the findings of this study are available from the corresponding author upon reasonable request.

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

## Acknowledgements

We thank A. Steffens, M. Kliesch and C. Ferrie for discussions and comments on the manuscript. This work has been supported by the Templeton Foundation, the EU (RAQUEL, AQuS), the ERC (TAQ), by the Freie Universität Berlin and the University of Cologne within the Excellence Initiative of the German Federal and State Governments, the DFG (SPP 1798 CoSIP, EI 519/7-1, EI 519/9-1, CRC 183 and GRO 4334/2-1), by the Austrian Science Fund (FWF) through the SFB FoQus (FWF Project No. F4002-N16), the Institut für Quanteninformation GmbH, the BMBF (Q.com), as well as by Universities Australia and DAAD's Joint Research Co-operation Scheme (using funds provided by the German Federal Ministry of Education and Research). This work was also supported by the Australian Research Council via EQuS project number CE11001013, and by the US Army Research Office under the grants W911NF-14-1-0098, W911NF-16-1-0070 and W911NF-14-1-0103 within the QCVV program. Furthermore, this research was funded by the Office of the Director of National Intelligence (ODNI), Intelligence Advanced Research Projects Activity (IARPA), through the Army Research Office grant W911NF-10-1-0284. All statements of fact, opinion or conclusions contained herein are those of the authors and should not be construed as representing the official views or policies of IARPA, the ODNI or the US Government. S.T.F. also acknowledges support from an Australian Research Council Future Fellowship FT130101744.

## Author contributions

C.A.R., D.G., S.T.F. and J.E. developed the theoretical framework. T.M., D.N. and R.B. designed and performed the experiment. C.A.R. carried out the data analysis and simulations. All authors contributed to the writing of the manuscript.

## Additional information

**Competing interests:** The authors declare no competing financial interests.

**Publisher's note**: 

