## [Peer review file · Nature Communications]

Reviewers' comments:

Reviewer #1 (Remarks to the Author):

The authors employ the tools of compressed sensing as applied to quantum tomography in order to diagnose the performance of an experiment that is meant to prepare pure states in a seven-qubit error correcting code. The main contributions of this work are: (1) They perform the first compressed sensing reconstructions of experimentally produced quantum states in a large dimensional Hilbert space; (2) They perform analysis to show how to use compressed sensing to recover the statistically significant information available in the data, which opens the possibility of efficiently diagnosing different types of errors in the experiment. In recent years, the authors have pioneered the field of compressed sensing tomography, and this is an interesting and important application of their work.

Contribution (1) is clear-cut and substantial. This is a first of its kind experiment with analysis based on compressed sensing.

In contribution (2) the authors argue, correctly, that in quantum state reconstruction, one should not fit to statistically “random” data. Thus, even though a state might have full rank, given the limited amount of data collected in the experiment, we have no prayer of faithfully reconstructing anything but the first few dominant eigenvectors and eigen values of the density matrix. Nor should we care to do so, for the goal of the experiment is to produce a pure state, and the long tail in the spectral decomposition beyond the dominant one or two eigenvectors provides no useful information about the actual imperfections in experiment. Applying compressed sensing techniques allows them to extract information associated with the relevant, statistically important eigenvectors, which may give insight into the nature of the errors that occurred in the experiment.

While I agree with this argument, I have a number of questions and requested clarifications.

- The authors employ three estimators to reconstruct the state, as shown in Figs. 1a,b,c. The first two are said to be limits of the “matrix lasso” given in Eq. (8). But in the supplemental material, two other estimators are given: least squares with a positivity constraint, and trace-norm minimization with a least squares and positivity constraint. These are intimately related, but not precisely connected in the manuscript. While least squared is certainly the same as matrix lasso with $\mu=0$, the trace-norm minimization is referred to loosely as “ $\mu \rightarrow \infty$.” For clarity the actual estimators used should be stated in the main text.

- If the actual estimator used for “ $\mu \rightarrow \infty$ ” is trace-norm minimization with a least squares and positivity constraint, how was the value of “epsilon” chosen in Eq. (15) of supplement? This value will have substantial impact on the bias of the estimator towards pure states.

- The GRAD algorithm is applied to efficiently solve least squares with positivity. How was the optimization in the trace minimization estimator carried out efficiently?

- The role of positivity in these estimators is obscured. The authors state in the main text that “The estimator does make sense for $\mu = 0$ if one adds the additional constraint that the

result be positive semi-definite.” The reason for this is that the measurement record is informationally complete for rank-1 states within the set of matrices in the positive semidefinite cone, as proven in a recent paper: A. Kalev et al., “Quantum tomography protocols with positivity are compressed sensing protocols,” npj Quantum Information 1, 15018 (2015). This should be appropriately referenced.

- Why was no trace constraint included in least squares estimator? A nature choice is $\text{Tr}(\rho)=1$. How would a different choice affect the bias in the estimator?

- The authors spend a substantial part of their manuscript discussing a spectral thresholding method that allows them to set to zero those eigenvalues of the density matrix that are statistically insignificant. However, the implication of this method is not clear. In particular, in Fig. 2c, the distribution of eigenvalues for this method is essentially indistinguishable from that of the least squares reconstruction. But this seems to indicate that the trace minimization, with their choice of epsilon, is chopping off statistically significant eigenvectors, while least squares with no trace constraint is capturing all the important data, and no more. This seems to be the opposite to the conclusion emphasized in the paper – that trace minimization captures the dominant eigenvectors and no more. What I am missing here?

- The authors argue that different estimators can help diagnose coherent vs. incoherent errors in the experiment. A similar conclusion was reached in Baldwin et al., “Quantum process tomography of unitary and near-unitary maps,” Phys. Rev. A 90, 012110 (2014), which should be referenced.

In summary, this is an interesting and important piece of work. I agree with the main thrust of the paper, and general thesis – that compressed sensing is a useful diagnostic for large dimensional systems, even if the rank is not small but dominated by a few eigenvalues. They have new techniques to distinguish statistically relevant information from “noise.” Their techniques can make compressed sensing tomography a useful diagnostic for large dimensional systems in cases where it would otherwise be intractable. However, I cannot recommend publication without clarification and correction on the points above.

Postscript – I noticed a few typos. Most substantially, right before the conclusion they state, “The results are shown in part (d) of Fig. 1.” I assume they mean part (c) of Fig. 1.

Reviewer #2 (Remarks to the Author):

Remarks to the authors

The paper titled "Experimental quantum compressed sensing for a seven-qubit system" by Riofrio and collaborators describes both new numerical and heuristic approaches to compressed sensing in quantum tomography, as well as their implementation on a seven-qubit ion trap system. The challenges they face are significant: the 7 qubit states they consider require many operations to prepare, and reconstructing such highly entangled states from experimental data is incredibly challenging. The authors techniques focus on the latter task: reconstructing the state of such a system in as efficient a manner as possible, and interpreting these results in a meaningful way.

###

The main experimental contribution is the implementation of compressed sensing on a relatively large quantum system : a system so large that traditional state tomography (via linear regression or maximum likelihood) would be prohibitively expensive, as it would require the repeated measurement of over 16k different experimental settings. Moreover, the state being reconstructed is encoded in a topological code of great promise for fault-tolerant quantum computation, and so the analysis of this experiment is highly relevant to the community. From the experimental perspective, there is no question that this is a very challenging, complex, and *necessary* experiment. In order to build large quantum computers, we will need to perform encoded computation, and we will need to find ways to debug such large systems. The authors aim to propose precisely one such debugging technique: low rank reconstructions of quantum systems.

The main theoretical contribution of this paper is an argument (supported by some theoretical and experimental evidence) for low rank reconstruction even when the state being reconstructed is not known to be low rank (i.e., close to "pure"). The argument seems to boil down to model selection: one should choose the most complex model that is well supported by the data, and avoid models that are more complex than that, as they end up fitting to statistical fluctuations. Determining the appropriate level of complexity for the model is no trivial task, and any technique that shows promise in correctly identifying the appropriate model complexity would be an important contribution.

This model selection argument is not by any means new, but the specific technique the authors propose to implement it ("spectral thresholding") is very intriguing, largely because some of the numerical evidence presented in the supporting material seems rather convincing. Fig 1-4 of the supplementary material, in particular, show that under numerical studies the matrix rank obtained by spectral thresholding coincides with the true rank of the state simulated — as long as enough data is taken. The theoretical heuristic for the derivation of spectral thresholding is also highly plausible and has a rather simple intuition behind it: if the fluctuations of the observations are compatible with random matrices, then those observations can be taken to be just "noise" and the model should ignore them.

Finally, the numerical approach to the tomographic estimator seems very attractive. The gradients within the physical space (positive semi-definite operators) can be computed

exactly with a compact expression, and once can expect that this leads to rather simple customized implementations of the estimators without the need for convex optimization packages (that although flexible, can be slow).

From these perspectives, there is no question that the paper has explored interesting approaches to relevant questions in the field of quantum information, and makes interesting contributions.

##

Despite these various contributions, however, there are a number of significant issues with the paper that I think must be addressed before it can be seriously considered for publication.

The first significant issue, in my opinion, is the great unexplained discrepancy between the matrix Lasso estimate (μ_{∞}) and either the constrained least squares ($\mu=0$) or the spectral thresholding reconstruction. Although a fidelity of ~ 0.3 in a seven qubit experiment is a non-trivial achievement, the gap between ~ 0.98 and ~ 0.3 in the reconstructions is astounding and needs to be better explained. It is hard to reconcile how such highly distinguishable reconstructions can both be consistent with a single experiment. The natural question is: which should be trust? The paper does not address this question, which strikes me as puzzling.

The authors themselves acknowledge to have no explanation for the gap, and the hypotheses proposed (that the matrix Lasso is much more heavily biased, and that the dataset collected was too limited) are not fully explored. One would expect at very least a numerical exploration of which of the two extremal fidelities is to be trusted, and which is too biased to be meaningful. Instead, the authors commit to the spectral thresholding technique and discuss the performance of the heuristics for this model selection technique, but neglect to make a similar analysis for the matrix LASSO.

The supplementary material (SM) *does* describe a numerical exploration of spectral thresholding, but once again it betrays the authors commitment to that technique — there is no comparison with the matrix LASSO to be found.

The numerical estimates of expected risk found in the SM partially addresses the question, as it says at least something about how spectral thresholding performs. The authors choice of the Frobenius norm as a risk measure, on the other hand, seems questionable. It would seem much more natural to consider the trace norm as a measure of risk, as it is closely connected with single-shot distinguishability. Although these norms can be used to bound each other, in a numerical study there seems to be little reason to choose a norm without an operational interpretation over a norm with one. Moreover, the plots in Fig 7 & 8 not do seem to yield any simple interpretation. For example, a small number of different settings seem to yield low risk estimates with low rank, but only if the true state has high rank -- an explanation of how biased low rank estimates can approximate high rank states well is lacking.

It is easy to imagine that the coarse pixelation used can heavily distort any relationship

between the quantities being studied in these plots. For that reason, a more careful study of risk under both the matrix lasso and spectral thresholding, for example, would be invaluable.

It should be emphasized that spectral thresholding seems to work remarkably well when there are enough measurement settings. The argument that it is desirable to underestimate the rank when few settings are used strikes me as questionable — the less one looks at one's experiment, the better it will look! The trade off is, of course, increased risk, which further motivates revisiting the risk analysis.

The argument for rank deficient reconstructions of states that are not rank deficient also encourages further comparison between the LASSO and spectral thresholding. It seems implicit that the flaw with the matrix LASSO is that it underestimates the rank much more heavily than spectral thresholding. For strictly unitary errors, however, there seems to be no reason to look at spectral thresholding — other than for the purpose of more easily certifying the unitarity of the error more easily — as unitary errors do not change the rank of the state.

Even if this approach to characterize unitary errors were to be taken, it would only apply to "atomic" unitary evolution, in the sense that it would be nearly hopeless to try to debug a quantum control protocol consisting of many non-commuting gates acting on different parts of the system. If one attempted to debug the unitary preparation of a topological code by looking at the overall unitary error (if it was indeed largely unitary), it would be essentially impossible to determine which of the many steps failed and how without additional assumptions or checks.

This, in turn, raises the question of non-unitary errors in the experimental data: since there is a large discrepancy between the LASSO estimate (which should account largely for unitary errors), and the other estimates (that are more sensitive to non-unitary, rank increasing errors), should we conclude the experiments are dominated by non-unitary errors? Any significant discussion of how the experiments should be interpreted is sorely lacking, and the potential connections are only hinted at.

##

Overall, the main flaw of the paper is that it is unclear what conclusion we should reach about either the experiment, or a choice between the matrix LASSO and spectral thresholding. Data was collected, it was processed in different reasonably well-motivated ways, and very different estimates were reached. Which is the one that best captures the accuracy of the experiments? What can be learned, even in principle, from these different approaches to tomographic reconstruction? That question is simply unaddressed in this paper.

One final and independent comment. Given that this paper describes the analysis of experimental data, it is only reasonable to include the experimental data and the code used to perform the analysis. I see no reasonable argument that would support withholding the data from public release. If the authors have reasons to withhold the code used in the analysis, detailed pseudo code should suffice. There is no question, however, that data, along with a sufficiently detailed description of the analysis performed on the data, is necessary for reproducibility of the results.

Conclusion

In conclusion, given the flaws in the analysis presented in the paper, I cannot recommend it for publication. I do not think these issues are fatal, but they do require significant rewriting and rethinking of the analysis presented in the paper.

Additional comments about specific passages

- Page 1

+ In the 3rd paragraph the authors claim that in compressed sensing "there is no need to make any a priori assumptions about the true quantum state". Immediately after, they write "Quantum compressed sensing is most effective on density matrices with quickly decaying eigenvalues". Although strictly correct (one can test the validity of the reconstructed test without needing to assume the state is low rank before reconstruction), this subtle point needs to be clarified more explicitly — the paper should not be addressing only the experts in compressed sensing, but the community at large.

- Page 2

+ I find the focus on *anticipated states* instead of *ideal states* rather odd. Surely one can anticipate states that are far from ideal if one knows the flaws of the experiment. The anticipated states discussed in the paper are indistinguishable from ideal states. So what is the meaning of the distinction?

- Page 3

+ "The estimator does make sense for $\mu = 0$ if one adds the additional constraint that the result be positive semidefinite". Should we take "does make sense" to mean "is consistent", in the strict statistical sense?

- Page 4

+ When the authors discuss Fig. 1, they mention "valid and faithful reconstructions of the anticipated state", but one reconstruction has $F=0.98$, while the others have $F\sim 0.3$. In what sense are both of these faithful with experiments if they are not faithful with each other?

- Page 6

+ The conclusion of the paragraph before "Conclusion and perspectives" points to "part (d) of Fig. 1". Neither Fig. 1 in the main body nor Fig. 1 in the SM have a part (d).

Reviewer #3 (Remarks to the Author):

The paper presents the largest-scale realization of quantum compressed sensing in a seven-qubit system implemented with trapped ions. The work addresses a roadblock in quantum information processing as the Hilbert space scales exponentially and thus quantum tomography will become only feasible when performed with an incomplete set of measurements. The work is thus of great importance to the interdisciplinary field of quantum information processing implemented by a broad range of physical systems ranging from superconducting qubits, trapped ions, solid state qubits and cold atoms. A new numerical method is presented to perform quantum compressed sensing. Additional supplementary material will allow other groups to implement their method.

The data is presented in six figures. Adequate details to statistics is given in figure caption and main text. In order to support other researchers to reproduce the work it would be helpful to provide additional supplementary material with code and data such that the analysis could be verified. I consider this crucial for other groups to be able to apply the presented method to their experiments as well for theory groups to improve on the algorithms using supplied data.

In conclusion I fully support publication of the paper in Nature Communications due to the high quality and novelty of the work. I hope the authors will provide additional supplementary material as suggested.

Reply to Reviewer #1

We would like to thank the reviewer for the careful and essentially very positive report. We are delighted to read that he or she thinks our work is substantial. At the same time, he or she adds remarks, comments and questions, all of which we have taken very seriously. Subsequently, we will detail what we have done in order to fully accommodate those comments.

The authors employ the tools of compressed sensing as applied to quantum tomography in order to diagnose the performance of an experiment that is meant to prepare pure states in a seven-qubit error correcting code. The main contributions of this work are: (1) They perform the first compressed sensing reconstructions of experimentally produced quantum states in a large dimensional Hilbert space; (2) They perform analysis to show how to use compressed sensing to recover the statistically significant information available in the data, which opens the possibility of efficiently diagnosing different types of errors in the experiment. In recent years, the authors have pioneered the field of compressed sensing tomography, and this is an interesting and important application of their work.

All this is correct, and we agree with the general assessment.

Contribution (1) is clear-cut and substantial. This is a first of its kind experiment with analysis based on compressed sensing.

In contribution (2) the authors argue, correctly, that in quantum state reconstruction, one should not fit to statistically “random” data. Thus, even though a state might have full rank, given the limited amount of data collected in the experiment, we have no prayer of faithfully reconstructing anything but the first few dominant eigenvectors and eigenvalues of the density matrix.

Again, we agree.

Nor should we care to do so, for the goal of the experiment is to produce a pure state, and the long tail in the spectral decomposition beyond the dominant one or two eigenvectors provides no useful information about the actual imperfections in experiment.

This is again correct and much in agreement with the spirit of our work.

Applying compressed sensing techniques allows them to extract information associated with the relevant, statistically important eigenvectors, which may give insight into the nature of the errors that occurred in the experiment.

Indeed.

While I agree with this argument, I have a number of questions and requested clarifications.

- The authors employ three estimators to reconstruct the state, as shown in Figs. 1a,b,c. The first two are said to be limits of the “matrix lasso” given in Eq. (8).

This is right - we put emphasis on them in the main text as these are of key importance.

But in the supplemental material, two other estimators are given: least squares with a positivity constraint, and trace-norm minimization with a least squares and positivity constraint. These are intimately related, but not precisely connected in the manuscript. While least squares is certainly the same as matrix lasso with $\mu=0$, the trace-norm minimization is referred to loosely as " $\mu \rightarrow \infty$." For clarity the actual estimators used should be stated in the main text.

This is a good point. We took this remark very seriously and altered both the main manuscript and the supplementary material accordingly. Indeed, it is very important that it is very clear what estimators we make use of. Because of the confusion this has apparently created, we have eliminated the discussion of the matrix Lasso estimator and instead discussed the trace norm minimizer (TNM) and least squares (LS) estimators separately in the main text. In retrospect and in the light of the reports of the referees, we can now see that this improves readability of the manuscript. We thank the referee for the comment.

- If the actual estimator used for " $\mu \rightarrow \infty$ " is trace-norm minimization with a least squares and positivity constraint, how was the value of "epsilon" chosen in Eq. (15) of supplement? This value will have substantial impact on the bias of the estimator towards pure states.

In the previous version of the manuscript, we have chosen a large enough epsilon compared with the expected statistics of the measurements. In this sense, the epsilon chosen was heavily biasing the reconstructed state towards a pure state. Since the statistics of the data are not so impressive (100 repetitions and 128 possible outcomes), it turns out we are in a regime far from seeing concentration effects, i.e., single shot observations are very far from the expected statistics. In that regime, one has 2 options: 1. choose epsilon large enough so almost pure states are reconstructed, or 2. choose epsilon minimal, i.e., the smallest value that makes the estimator converge. We originally presented the former, hoping that we could make a clear point on the impact of the choice of the estimator would have on the estimate, but in retrospect realise that this discussion was rather confusing than clarifying our main point.

In our revisited version of the manuscript, we have hence decided that it is better to make the second choice instead of the first and have made that change throughout the text. We observe that this choice gives more homogenous performance among the estimators used, while still being compatible with the statistically significant spectrum our model selection protocol suggest keeping. It also gives a very clear advice what to do, as our model selection procedure gives a clear justification of the rank one should consider for the estimate.

- The GRAD algorithm is applied to efficiently solve least squares with positivity. How was the optimization in the trace minimization estimator carried out efficiently?

When we say "efficient" we mean efficient in the quantum system's dimension. In this sense, interior point methods for semidefinite programs are used in the trace-minimization estimator. These methods have a polynomial runtime in the dimension of the Hilbert space (in fact, better than cubic). Of course, no method for quantum state reconstruction can be efficient in the number of particles since the dimension of the Hilbert space scales exponentially with the number of constituents. That being said, there are many "efficient" algorithms for TNM, such as singular value thresholding. In our case, it was not needed to use them, but if we had used them, we would have obtained similar results. It is key to have

feasible reconstruction methods for intermediate sized quantum systems available. We hence argue that this method development is important in its own right.

- The role of positivity in these estimators is obscured. The authors state in the main text that “The estimator does make sense for $\mu = 0$ if one adds the additional constraint that the result be positive semi-definite.” The reason for this is that the measurement record is informationally complete for rank-1 states within the set of matrices in the positive semidefinite cone, as proven in a recent paper: A. Kalev et al., “Quantum tomography protocols with positivity are compressed sensing protocols,” *npj Quantum Information* 1, 15018 (2015). This should be appropriately referenced.

We have included this reference and commented upon the role of positivity. We are grateful to the referee for pointing this out. We now write “In fact, the additional positivity constraint in this type of problems renders essentially all estimators equivalent” and cite this work.

- Why was no trace constraint included in least squares estimator? A nature choice is $\text{Tr}(\rho)=1$. How would a different choice affect the bias in the estimator?

The least squares estimator is realized via the GRAD algorithm. For simplicity, we do not enforce the trace constraint explicitly, since, numerically, it is preferable to solve an unconstrained optimization. For this reason, we leave the trace as a free variable and renormalize at the end. A different choice of normalization would definitely affect the final estimated state. Although not in the context of this project, we have looked in detail at this question and have empirically tested several normalization methods: 1) Dividing the resulting matrix, X , by its trace, 2) adding to each non-zero eigenvalue of the resulting matrix, X , $\text{trace}(X)/r$, where $r=\text{rank}(X)$. 3) Solving an additional, positive least squares problem in which the eigenvalues of the resulting matrix, X , are rescaled appropriately via a weighted optimization that takes into account the measured operators. Empirically, via simulated data, we have found that the first method produces estimates closer to the true state when pure states are considered. Hence, we are confident that the approach taken in this work is meaningful.

- The authors spend a substantial part of their manuscript discussing a spectral thresholding method that allows them to set to zero those eigenvalues of the density matrix that are statistically insignificant. However, the implication of this method is not clear. In particular, in Fig. 2c, the distribution of eigenvalues for this method is essentially indistinguishable from that of the least squares reconstruction. But this seems to indicate that the trace minimization, with their choice of epsilon, is chopping off statistically significant eigenvectors, while least squares with no trace constraint is capturing all the important data, and no more. This seems to be the opposite to the conclusion emphasized in the paper – that trace minimization captures the dominant eigenvectors and no more. What I am missing here?

We again agree with the referee. Consequently, we have modified the manuscript to make this point considerably clearer. We know that TNM has the potential to bias the reconstructed state towards pure states. The amount of bias depends on the parameter ϵ , as discussed above.

Our heuristic model selection technique, in contrast, gives rise to a clear advice what to do. By using a different estimator, in our case LS, we intended to understand how one would truncate it into a low-rank state and whether that estimate would be close to what

TNM produces. For this purpose, we developed a model selection heuristic to decide which part of the spectrum of ρ is statistically relevant. In the previous version of the manuscript, we presented the results for large ϵ , as already mentioned above. We realized now that it would rather confuse the reader, and that we should stick to a value for ϵ that would be compatible with our model selection procedure. We have corrected this error and carefully modified the text to clarify this point. We again thank the referee for pointing this out to us. We are now much happier with this leaner and cleaner version of the manuscript.

- The authors argue that different estimators can help diagnose coherent vs. incoherent errors in the experiment. A similar conclusion was reached in Baldwin et al., "Quantum process tomography of unitary and near-unitary maps," Phys. Rev. A 90, 012110 (2014), which should be referenced.

This is right and we reference this work in the new version and write "in the context of process tomography of near unitary maps, a similar effect has been observed".

In summary, this is an interesting and important piece of work.

Thanks a lot again. We do appreciate this assessment.

I agree with the main thrust of the paper, and general thesis – that compressed sensing is a useful diagnostic for large dimensional systems, even if the rank is not small but dominated by a few eigenvalues. They have new techniques to distinguish statistically relevant information from "noise." Their techniques can make compressed sensing tomography a useful diagnostic for large dimensional systems in cases where it would otherwise be intractable. However, I cannot recommend publication without clarification and correction on the points above.

Postscript – I noticed a few typos. Most substantially, right before the conclusion they state, "The results are shown in part (d) of Fig. 1." I assume they mean part (c) of Fig. 1.

We have corrected this and also a few other typos. Again, we thank the reviewer for the careful, thoughtful and positive report. We hope that having carefully accommodated all remarks, the manuscript is now suitable for publication.

Reviewer #2 (Remarks to the Author):

Remarks to the authors

The paper titled "Experimental quantum compressed sensing for a seven-qubit system" by Riofrio and collaborators describes both new numerical and heuristic approaches to compressed sensing in quantum tomography, as well as their implementation on a seven-qubit ion trap system. The challenges they face are significant: the 7 qubit states they consider require many operations to prepare, and reconstructing such highly entangled states from experimental data is incredibly challenging. The authors techniques focus on the latter task: reconstructing the state of such a system in as efficient a manner as possible, and interpreting these results in a meaningful way.

We would like to thank the referee very much for the careful report. We appreciate the referee's clear understanding of the challenge involved in this endeavor and also our point in transforming these techniques into practical and usable tools.

The main experimental contribution is the implementation of compressed sensing on a relatively large quantum system : a system so large that traditional state tomography (via linear regression or maximum likelihood) would be prohibitively expensive, as it would require the repeated measurement of over 16k different experimental settings. Moreover, the state being reconstructed is encoded in a topological code of great promise for fault-tolerant quantum computation, and so the analysis of this experiment is highly relevant to the community. From the experimental perspective, there is no question that this is a very challenging, complex, and *necessary* experiment. In order to build large quantum computers, we will need to perform encoded computation, and we will need to find ways to debug such large systems. The authors aim to propose precisely one such debugging technique: low rank reconstructions of quantum systems.

We again thank the referee for his/her careful assessment and understanding of the necessity of practical system identification techniques and the experiment.

The main theoretical contribution of this paper is an argument (supported by some theoretical and experimental evidence) for low rank reconstruction even when the state being reconstructed is not known to be low rank (i.e., close to "pure"). The argument seems to boil down to model selection: one should choose the most complex model that is well supported by the data, and avoid models that are more complex than that, as they end up fitting to statistical fluctuations. Determining the appropriate level of complexity for the model is no trivial task, and any technique that shows promise in correctly identifying the appropriate model complexity would be an important contribution.

Yes, we again agree with the referee.

This model selection argument is not by any means new, but the specific technique the authors propose to implement it ("spectral thresholding") is very intriguing, largely because some of the numerical evidence presented in the supporting material seems rather convincing.

Indeed, we again agree. It is not new to make use of model selection techniques as such. The way we implement it is the intriguing aspect, and we think that we are presenting a feasible and practical approach here for intermediate sized quantum systems.

Fig 1-4 of the supplementary material, in particular, show that under numerical studies the matrix rank obtained by spectral thresholding coincides with the true rank of the state simulated — as long as enough data is taken. The theoretical heuristic for the derivation of spectral thresholding is also highly plausible and has a rather simple intuition behind it: if the fluctuations of the observations are compatible with random matrices, then those observations can be taken to be just "noise" and the model should ignore them.

Indeed, the numerical evidence presented in the supplemental material supports these claims.

Finally, the numerical approach to the tomographic estimator seems very attractive. The gradients within the physical space (positive semi-definite operators) can be computed exactly with a compact expression, and one can expect that this leads to rather simple customized implementations of the estimators without the need for convex optimization packages (that although flexible, can be slow).

We again agree with the referee. Our gradient based method is extremely simple and efficient.

From these perspectives, there is no question that the paper has explored interesting approaches to relevant questions in the field of quantum information, and makes interesting contributions.

We thank the referee for this comments.

Despite these various contributions, however, there are a number of significant issues with the paper that I think must be addressed before it can be seriously considered for publication.

Thanks again for this.

The first significant issue, in my opinion, is the great unexplained discrepancy between the matrix Lasso estimate ($\mu \rightarrow \infty$) and either the constrained least squares ($\mu=0$) or the spectral thresholding reconstruction. Although a fidelity of ~ 0.3 in a seven qubit experiment is a non-trivial achievement, the gap between ~ 0.98 and ~ 0.3 in the reconstructions is astounding and needs to be better explained. It is hard to reconcile how such highly distinguishable reconstructions can both be consistent with a single experiment. The natural question is: which should be trusted? The paper does not address this question, which strikes me as puzzling.

The authors themselves acknowledge to have no explanation for the gap, and the hypotheses proposed (that the matrix Lasso is much more heavily biased, and that the dataset collected was too limited) are not fully explored. One would expect at very least a numerical exploration of which of the two extremal fidelities is to be trusted, and which is too biased to be meaningful. Instead, the authors commit to the spectral thresholding technique and discuss the performance of the heuristics for this model selection technique, but neglect to make a similar analysis for the matrix LASSO.

We understand the confusion generated by these fidelity values and we have modified the manuscript to better explain them. The situation is as follows: When highly incomplete data are available, there is simply no way to determine all parameters needed to describe

a unique quantum state. This is why we developed our heuristic model selection procedure, to have a systematic guidance of how to proceed.

Notably, different estimators will implicitly or explicitly amount to making different assumptions; for example, as described now much clearer in the text, our TNM estimator will produce either very mixed or very pure reconstructed states depending on the error level ϵ . Other estimators, for example, MLE or LS will produce more mixed states in the low data regime. This should not be a surprise, however, since there is an infinite number of valid states that are compatible with incomplete data. It is just that different estimators will find different states.

The core question, needless to say, is the one “which should be trust?”. We make the distinction between “incoherent” and “coherent” experimental errors more clear in the manuscript. We argue that coherent errors are better seen if one biases the reconstructed state towards pure states. There, one could clearly see whether unwanted superpositions are created in the experiment, for example. If, however, one is more interested in incoherent errors, one would want to see the shape of the tail of the spectrum of the reconstructed state, which may consist mainly of noise.

To avoid any confusion, and to have a more consistent and coherent presentation, we have modified the plots and presented the results of a much less biased TNM estimate in figure 1. Now the fidelity is not as different from the other estimators because all the estimators give more mixed states as a result, and all presented data are very much compatible with our model selection procedure.

However, we have added a paragraph in the supplemental material in which we carefully discuss what happens when we bias the TNM estimator towards pure states and added a new plot showing the high fidelity estimates we presented in the first version of the manuscript. Note that when reconstructing more pure states, the overlap with the anticipated states is much higher and thus the fidelity is also much higher. We hope the modifications and new discussion improve the readability of the paper.

The supplementary material (SM) *does* describe a numerical exploration of spectral thresholding, but once again it betrays the author's commitment to that technique — there is no comparison with the matrix LASSO to be found.

We have taken this comment as an invitation to be clearer in this respect in the presentation of the supplementary material. Obviously, it is by no means our aim to make the impression that we take anything for granted. In contrast, the role of the supplements is to give the interested reader the chance to go through the fine print of our methods in all detail and clarity. We have the impression that the referee may have misunderstood the purpose of this analysis, which we have now presented in clearer terms.

The purpose of this section is to exemplify the significance and strength of our heuristic model selection to give good advice about the rank of the object to be reconstructed. Specifically, it is about being able to detect the true rank. For this reason, in this assessment, we did not consider it necessary to make this comparison (previously LASSO, now with the TNM estimator). The purpose of the simulations was to make a comparison with the actual rank of the true state. Making the comparison to a reconstructed state via a different method would not shed light on the validity of the heuristic since the rank of the reconstructed state will greatly vary with different parameter choices of the estimator. Again: The LASSO and TNM are estimators, the spectral thresholding is a technique to identify

the most suitable rank. In the new version, we have emphasised the purpose of this analysis in clearer terms.

The numerical estimates of expected risk found in the SM partially addresses the question, as it says at least something about how spectral thresholding performs. The authors choice of the Frobenius norm as a risk measure, on the other hand, seems questionable. It would seem much more natural to consider the trace norm as a measure of risk, as it is closely connected with single-shot distinguishability. Although these norms can be used to bound each other, in a numerical study there seems to be little reason to choose a norm without an operational interpretation over a norm with one.

We acknowledge the point of the referee. The reason for this choice is that the Frobenius norm is common in risk assessments of such a type in this context. At the same time, there is the operational interpretation that the referee refers to. At the end, it is a bit a matter of taste, and there is an element of randomness in the choice. Fortunately, as the referee points out, one norm can be bound by the other and thus the choice does not alter our observation. If the referee insists, however, we are perfectly happy to change that.

Moreover, the plots in Fig 7 & 8 not do seem to yield any simple interpretation. For example, a small number of different settings seem to yield low risk estimates with low rank, but only if the true state has high rank -- an explanation of how biased low rank estimates can approximate high rank states well is lacking.

It is easy to imagine that the coarse pixelation used can heavily distort any relationship between the quantities being studied in these plots. For that reason, a more careful study of risk under both the matrix lasso and spectral thresholding, for example, would be invaluable.

We have explained better what we did here. Fig. 7 and 8 are intended as a comparison between our thresholding technique and a different one from ref. 5 of the supplemental material. We have made it clearer what the aim of this comparison is. The referee is correct that those plots do not give rise to a clear advice, which in itself is an evidence that both methods produce similar results, even though they are conceptually very different. Again, we have not added a comparison between matrix LASSO and spectral thresholding, as the two methods serve a different purpose, as explained above and also made clearer in the version of the manuscript. We hope this has now clarified the points raised.

It should be emphasized that spectral thresholding seems to work remarkably well when there are enough measurement settings. The argument that it is desirable to underestimate the rank when few settings are used strikes me as questionable — the less one looks at one's experiment, the better it will look! The trade off is, of course, increased risk, which further motivates revisiting the risk analysis.

This is an interesting point, and we have two things to say about this. 1. A fair and honest outcome of any tomography analysis cannot be the raw plot alone, but must be the plot in conjunction with a clear specification of what is shown. This is basically the “caption”, the explanation of what is provided with what statistical significance. 2. The “true” state is not known in any tomography experiment, and at best one can compare the reconstructed state with the “anticipated state”, which, however, may or may not have been prepared in the laboratory. It could be that the state is low rank, but not unit rank, and then the spectral thresholding method will give good advice what to give back.

As indeed, if the anticipated state is pure, but for some reason, the actually prepared state does have a rank larger than unity, then the reconstruction should indeed estimate the higher rank. If it cannot, for not enough data being available, all there is to be done is to find the simplest explanation of the outcomes, and say so very clearly. This is what one should say, as there is nothing one can say otherwise with any significance or justification, completely ignoring what we called the “caption” above.

The argument for rank deficient reconstructions of states that are not rank deficient also encourages further comparison between the LASSO and spectral thresholding. It seems implicit that the flaw with the matrix LASSO is that it underestimate the rank much more heavily than spectral thresholding. For strictly unitary errors, however, there seems to be no reason to look at spectral thresholding — other than for the purpose of more easily certifying the unitarity of the error more easily — as unitary errors do not change the rank of the state.

We have clarified this discussion in the main text. In our new discussion, we have two estimators that replace the LASSO, as we think it helps with the readability of the paper and transport a more coherent main message. In this new setting, the TNM is the estimator that bias the reconstructed states towards pure states, depending on the error level ϵ . In this sense, there is no intrinsic “flaw” in any of the estimators we use in this work. They perform differently in different situations. Our objective is to showcase this fact and raise awareness.

Even if this approach to characterize unitary errors were to be taken, it would only apply to “atomic” unitary evolution, in the sense that it would be nearly hopeless to try to debug a quantum control protocol consisting of many non-commuting gates acting on different parts of the system. If one attempted to debug the unitary preparation of a topological code by looking at the overall unitary error (if it was indeed largely unitary), it would be essentially impossible to determine which of the many steps failed and how without additional assumptions or checks.

The referee is right that trying to debug the overall error of a composite unitary only looking at the complete action over the system is nearly impossible for large enough systems. However, in practice, as the referee points out, one would never attempt doing such thing. One would certify each gate separately and make sure each simpler gate is working properly. In this sense, our procedure is still a valid “debugging” tool.

This, in turns, raises the question of non-unitary errors in the experimental data: since there is a large discrepancy between the LASSO estimate (which should account largely for unitary errors), and the other estimates (that are more sensitive to non-unitary, rank increasing errors), should we conclude the experiments are dominated by non-unitary errors? Any significant discussion of how the experiments should be interpreted is sorely lacking, and the potential connections are only hinted at.

Indeed, this is again a good point. We thank the referee for pointing out this omission. We have corrected this in the current version. From our analysis and our current understanding of the different strengths of the estimators used, we see that the data reveals mostly incoherent errors since the principal components of the density matrix are in excellent agreement with the anticipated states. We now discuss this in detail in the new version of the manuscript. We would like to remark that observing this behaviour purely by luck in such a large Hilbert space is nearly impossible.

##

Overall, the main flaw of the paper is that it is unclear what conclusion we should reach about either the experiment, or a choice between the matrix LASSO and spectral thresholding. Data was collected, it was processed in different reasonably well-motivated ways, and very different estimates were reached. Which is the one that best captures the accuracy of the experiments? What can be learned, even in principle, from these different approaches to tomographic reconstruction? That question is simply unaddressed in this paper.

We have carefully addressed the concerns of the referee in the new version of the paper. As explained above, we have added a comment on the differences of the estimators in the paper and clarified our main point. We now state more clearly, to avoid any misunderstanding, that the spectral thresholding is no estimator as such, but a technique to achieve model selection in conjunction with an estimator. We also state more clearly how the findings from tomography give rise to actionable advice in experiments. The presentation of the main text has become leaner and clearer, while all details are still available in the supplementary material. We agree with the referee that the previous version of the manuscript was lacking clarity and we have done our very best to fix this.

One final and independent comment. Given that this paper describes the analysis of experimental data, it is only reasonable to include the experimental data and the code used to perform the analysis. I see no reasonable argument that would support withholding the data from public release. If the authors have reasons to withhold the code used in the analysis, detailed pseudo code should suffice. There is no question, however, that data, along with a sufficiently detailed description of the analysis performed on the data, is necessary for reproducibility of the results.

This is a very good point and we very much share this sentiment. We will surely make the code public, for precisely the reasons the referee mentions. How this can be technically done will be decided by the editors. Either it can be part of the supplements, or we would, depending on the wishes of the journal, make it available elsewhere.

Conclusion

In conclusion, given the flaws in the analysis presented in the paper, I cannot recommend it for publication. I do not think these issues are fatal, but they do require significant rewriting and rethinking of the analysis presented in the paper.

We have carefully and seriously considered all the comments and have carefully accommodated all of them. We hope the referee considers the new manuscript suitable for publication in the present form.

Additional comments about specific passages

- Page 1

+ In the 3rd paragraph the authors claim that in compressed sensing "there is no need to make any a priori assumptions about the true quantum state". Immediately after, they write "Quantum compressed sensing is most effective on density matrices with quickly decaying eigenvalues". Although strictly correct (one can test the validity of the recon-

structed test without needing to assume the state is low rank before reconstruction), this subtle point needs to be clarified more explicitly — the paper should not be addressing only the experts in compressed sensing, but the community at large.

We have clarified this point.

- Page 2

+ I find the focus on *anticipated states* instead of *ideal states* rather odd. Surely one can anticipate states that are far from ideal if one knows the flaws of the experiment. The anticipated states discussed in the paper are indistinguishable from ideal states. So what is the meaning of the distinction?

The referee is right when noting that there is no distinction between the anticipated and the ideal states. For us, it is only a matter of preference in the terms. We like the word "anticipated" to make clear that this is what we expect the quantum machine, i.e., the experimental setup, to produce.

- Page 3

+ "The estimator does make sense for $\mu = 0$ if one adds the additional constraint that the result be positive semidefinite". Should we take "does make sense" to mean "is consistent", in the strict statistical sense?

Yes, thanks. We have modified the estimator section and removed all the confusing parts. We believe that the new version is much more clear.

- Page 4

+ When the authors discuss Fig. 1, they mention "valid and faithful reconstructions of the anticipated state", but one reconstruction has $F=0.98$, while the others have $F\sim 0.3$. In what sense are both of these faithful with experiments if they are not faithful with each other?

Please see the explanation above.

- Page 6

+ The conclusion of the paragraph before "Conclusion and perspectives" points to "part (d) of Fig. 1". Neither Fig. 1 in the main body nor Fig. 1 in the SM have a part (d).

We thank the referee for pointing this out and have corrected this mistake. Again, we thank the referee for the careful report.

Reviewer #3 (Remarks to the Author):

The paper presents the largest-scale realization of quantum compressed sensing in a seven-qubit system implemented with trapped ions. The work addresses a roadblock in quantum information processing as the Hilbert space scales exponentially and thus quantum tomography will become only feasible when performed with an incomplete set of measurements. The work is thus of great importance to the interdisciplinary field of quantum information processing implemented by a broad range of physical systems ranging from superconducting qubits, trapped ions, solid state qubits and cold atoms. A new numerical method is presented to perform quantum compressed sensing. Additional supplementary material will allow other groups to implement their method.

We thank the reviewer for the positive report. We are delighted to see that he or she thinks of our work being “of great importance to the interdisciplinary field of quantum information processing”. We appreciated this assessment.

The data is presented in six figures. Adequate details to statistics is given in figure caption and main text. In order to support other researchers to reproduce the work it would be helpful to provide additional supplementary material with code and data such that the analysis could be verified.

This is a very good point and we very much share this sentiment. We will surely make the code public, for precisely the reasons the referee mentions. How this can be technically done will be decided by the editors. Either it can be part of the supplements, or we would, depending on the wishes of the journal, make it available elsewhere.

I consider this crucial for other groups to be able to apply the presented method to their experiments as well for theory groups to improve on the algorithms using supplied data.

Again, thanks for making this point.

In conclusion I fully support publication of the paper in Nature Communications due to the high quality and novelty of the work. I hope the authors will provide additional supplementary material as suggested.

We are very happy to read that the referee is so enthusiastic about our work. Again, we thank for the careful and positive report.

REVIEWERS' COMMENTS:

Reviewer #1 (Remarks to the Author):

The authors have substantially revised their manuscript to address the main point of confusion -- the difference in the performance of the estimators: matrix-lasso, trace minimization, and least squares. There are many questions that remain as to how best to employ tomography, interpret the results, and use them to diagnose experimentally relevant errors. This manuscript makes important steps forward in that regard and is a useful contribution to the community as we develop more complex quantum information processing devices. I recommend publication in Nature Communications.

Reviewer #2 (Remarks to the Author):

I thank the authors for the changes and clarifications to the manuscript, and for their willingness to make the data and analysis software public.

Re: my request to compare spectral thresholding to other estimators such as the Lasso, I understand that spectral threshold is a heuristic to estimate the rank, not a tomographic estimator. However, in the text spectral thresholding it is used in conjunction with the LS optimization to create a new estimator by truncating the spectrum of the LS estimate (and in that way avoid overfitting).

The Lasso estimator usually is used in conjunction with cross-validation to fix the L1 penalty weight, and in that sense it is not all that different from the bootstrapped hypothesis test the authors propose for spectral thresholding. One can imagine the same cross-validation approach being used to fix the ϵ parameters of the TNM estimator. Clearly, the authors intend to use ϵ in different ways, so I will not insist on the comparison as I originally requested and explain above -- but I still hold it would be interesting to compare the performance of these different estimators. Perhaps the authors can reconsider making this addition.

Finally, in light of the improvements made to the manuscript, I have no objections to the acceptance of the paper into Nature Communications, and recommend its publication without the need for further review.